# Unison: Benchmarking Unified Multimodal Models via Synergistic Understanding and Generation

**Jinyu Liu** [1 2] **Xincheng Shuai** [3] **Henghui Ding** [3] **Yu-Gang Jiang** [1 2]

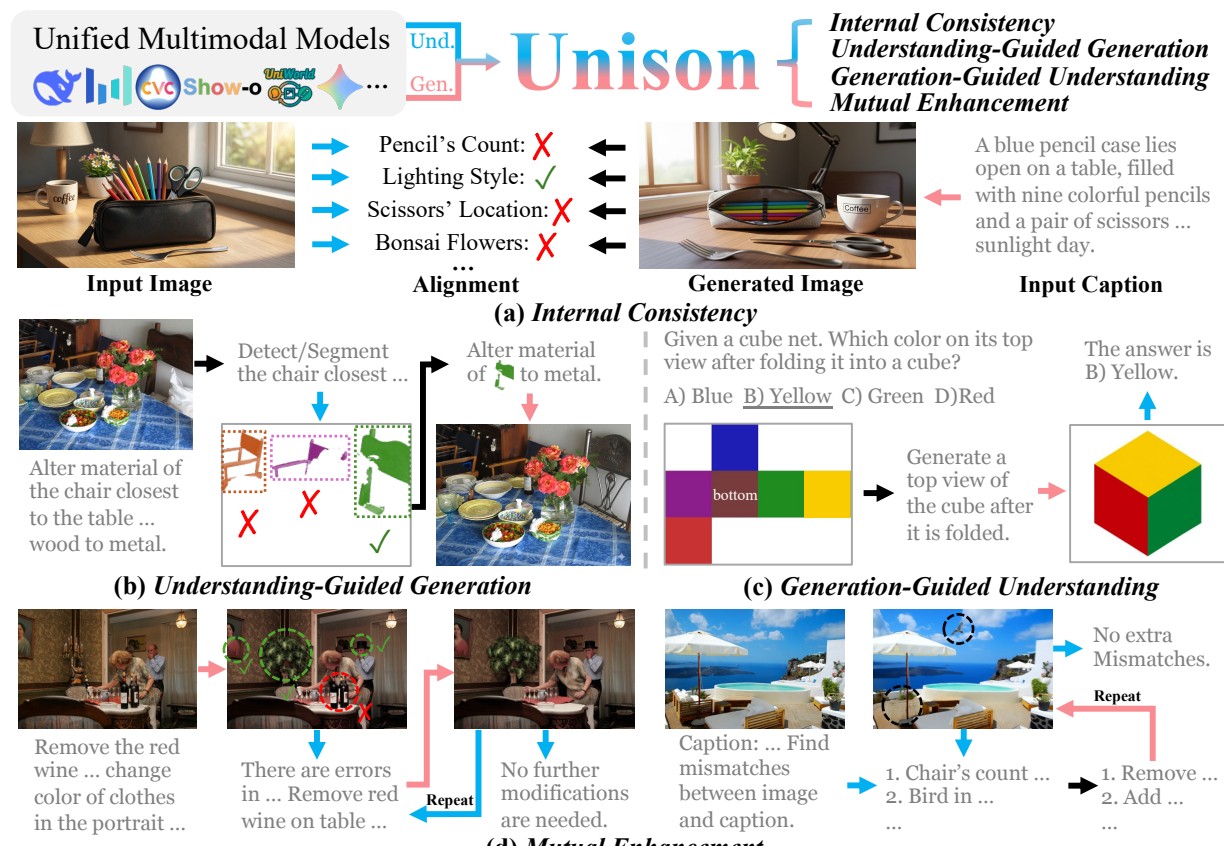

*Figure 1.* **Overview of Unison**. Unison decomposes unified models' capabilities into four dimensions: **(a)** *Internal Consistency* assesses internal alignment between understanding and generation. **(b)** *Und.-Guided Gen.* estimates model's ability to leverage comprehension guiding contextually generation. **(c)** *Gen.-Guided Und.* focuses on how synthesized outputs assist understanding tasks, and **(d)** *Mutual Enhancement* evaluates the iterative synergy where understanding identifies generation errors and vice versa, enabling mutual refinement.

## Abstract

Unified multimodal models capable of both understanding and generation have achieved remark-

[1]Institute of Trustworthy Embodied AI, Fudan University, Shanghai, China [2]Shanghai Key Laboratory of Multimodal Embodied AI, Shanghai, China [3]Institute of Big Data, College of Computer Science and Artificial Intelligence, Fudan University, Shanghai, China. Correspondence to: Yu-Gang Jiang <ygj@fudan.edu.cn>.

*Proceedings of the 43rd International Conference on Machine Learning*, Seoul, South Korea. PMLR 306, 2026. Copyright 2026 by the author(s).

able strides. However, despite their unified designs, existing evaluations typically assess understanding and generation capabilities in isolation, overlooking the synergy between comprehension and generation. To bridge this gap, we introduce **Unison**, a comprehensive benchmark comprising 2,169 high-quality unified task samples, designed to evaluate joint understanding and generation in unified multimodal models. Unison offers three key strengths: **1) Comprehensive Dimensions**: Unison encompasses *internal consistency*, *understanding-guided genera-*

*tion*, *generation-guided understanding*, and *mutual enhancement* to enable holistic evaluation. **2) Diagnostic Evaluation**: it provides both unified and decoupled tracks for understanding and generation, allowing fine-grained attribution of failure modes and quantitative analysis of the gains from unified modeling. **3) Human Alignment**: we also introduce Unison-Judge, an evaluation model well aligned with human judgments to ensure reliable assessment. Based on systematic evaluations of state-of-the-art models on Unison, we uncover critical limitations in current unified multimodal systems and highlight promising directions for future research. Unison will be publicly available at https://github.com/FudanCVL/Unison.

# 1. Introduction

Multimodal large language models (MLLMs) have advanced rapidly in recent years, demonstrating impressive comprehension capabilities in complex reasoning tasks. On the other hand, visual generative models, particularly diffusion-based, have achieved unprecedented fidelity and controllability in synthesizing images and videos. Motivated by their success, unified multimodal models (UMMs) have emerged as a rapidly advancing research direction, which aim to integrate complementary understanding (textual output) and generation (visual output) capabilities within a single framework. Current UMMs are increasingly moving beyond composition of separate understanding and generative models toward natively unified designs that support shared representations across both perception and synthesis.

Nowadays, UMMs have demonstrated strong performance in multimodal understanding, exhibiting notable complex reasoning capabilities on benchmarks such as MMBench (Liu et al., 2024) and MMMU (Yue et al., 2024). They also achieve compelling results in compositional image generation, producing fine-grained, instruction-aligned outputs, as evaluated on GenEval (Ghosh et al., 2023), DPG-Bench (Hu et al., 2024) and OneIG-Bench (Chang et al., 2025). However, existing evaluations remain confined to capability-specific tasks. There is a critical need for a benchmark to holistically evaluate UMMs on unified tasks requiring both understanding and generation capabilities.

To this end, we propose **Unison**, a comprehensive unified benchmark designed to holistically evaluate UMMs through synergistic understanding and generation. Our work is motivated by three foundational questions: *1) Do current UMMs truly integrate understanding and generation capabilities?* Prior work (Li et al., 2025a) points out that most UMMs suffer from a performance trade-off between these capabilities due to conflicting visual tokenization strategies. Uni-

Gen (Tian et al., 2025) leverage understanding ability as a self-verifier to assess the quality of generated images by evaluating image-text semantic coherence, suggesting a critical insight: If a UMM correctly comprehends an input image, can it generate a new image consistent with its own semantic interpretation? We define this notion as ***internal consistency***, as shown in Figure 1 (a), to evaluate the alignment between understanding and generation. *2) For challenging tasks, what is the synergistic effect of UMMs in leveraging their own reasoning to guide generation, and vice versa?* Bagel (Deng et al., 2025) claims that reasoning before visual generation may enhance the final output in multimodal tasks. By using this approach to filter training data, Bagel achieves superior performance. Motivated by this, we design ***understanding-guided generation*** and ***generation-guided understanding***, as shown in Figure 1 (b) and (c), to evaluate the synergistic effect in UMMs. *3) Can these dual capabilities synergistically interact to enhance the performance of generation or understanding tasks?* X-Omini (Geng et al., 2025) shows that shared autoregressive modeling between image and text generation, augmented by reward models, reduces cumulative errors, demonstrating that understanding and generation can mutually reinforce each other. Thus, we propose ***mutual enhancement*** to assess UMMs' self-refinement capability through iterative cycles of alternating multimodal understanding and image generation, as shown in Figure 1 (d).

Through the Unison framework, we observe that while certain UMMs achieve state-of-the-art performance on specialized understanding or generation benchmarks, they often fall short in unified tasks requiring joint comprehension and synthesis. Subsequent experiments and analysis further reveal key limitations of current UMMs and point to promising directions for future research.

In summary, our main contributions are as follows:

- In this work, we explore a promising yet under-researched space for evaluating unified multimodal models via synergistic understanding and generation. Based on this, we introduce Unsion, consisting of 2,169 human-validated unified task samples, which encompasses *internal consistency*, *understanding-guided generation*, *generation-guided understanding*, and *mutual enhancement* to enable unified evaluation.

- Unison supports both joint and decoupled assessment of understanding and generation within the same framework, providing interpretable and fine-grained analysis.

- Extensive experiments demonstrate that Unison-Judge exhibits strong alignment with human judgments. Additionally, we provide valuable insights into current UMMs' limitations and future directions.

## 2. Related works

**Unified Multimodal Models**. Recent years have witnessed remarkable advancement in both multimodal large language models (MLLMs) and visual generative models, leading to increasing research efforts toward unifying these capabilities within a single architecture. This pursuit has given rise to unified multimodal models (UMMs) (Chen et al., 2025b; Li et al., 2025b; Wu et al., 2025d; Xie et al., 2024; 2025; Ge et al., 2024; Xiao et al., 2025; Chen et al., 2025a; Deng et al., 2025; Huang et al., 2025; Wu et al., 2025c; Qu et al., 2025; Zhou et al., 2025; Lin et al., 2025), which generally fall into three categories: auto-regressive models, diffusion-based models and hybrid models. Auto-regressive models, such as SEED-X (Ge et al., 2024) and UniWorld (Lin et al., 2025), typically tokenize both visual and text inputs into discrete sequential tokens using transformer-based architectures, enabling seamless multimodal reasoning and generation. On the other hand, diffusion-based unified models such as D-DiT (Li et al., 2025b), which introduces a dual-branch diffusion framework that jointly generates text and images, facilitating coherent and controllable synthesis of aligned image-text pairs starting from pure noise, enabling synchronized generation across visual and textual modality. In addition, hybrid approaches (Xie et al., 2024; Zhou et al., 2025; Deng et al., 2025) integrate auto-regressive and diffusion-based components within a unified framework. These methods aim to preserve strong language reasoning while enabling high-fidelity visual generation, but they often increase training complexity and computational cost.

**Multimodal Understanding Benchmarks** have evolved from basic perception toward complex reasoning. Early datasets such as Flickr30k (Plummer et al., 2015) and MS COCO Captions (Chen et al., 2015) focused on image–text alignment, while VQA-style benchmarks (Antol et al., 2015; Goyal et al., 2017; Singh et al., 2019) introduced free-form question answering requiring scene-level comprehension. More recent works evaluate diverse multimodal capabilities: MMBench (Liu et al., 2024) emphasizes cross-lingual perception, MMMU (Yue et al., 2024) targets domain-specific reasoning, and SEED-Bench (Li et al., 2024) assesses fine-grained skills with external knowledge integration.

**Image Generation & Editing Benchmarks**. Previous image generation relied on automatic metrics like Inception Score (IS) (Salimans et al., 2016) for basic image-only quality assessment. Frechet Inception Distance (FID) (Heusel et al., 2017) and CLIPScore (Hessel et al., 2021) estimates image-text alignment in feature level. However, contemporary works (Ghosh et al., 2023; Huang et al., 2023; Hu et al., 2024) reveal significant gaps between these metrics and human judgment, particularly for complex scenes. To address this, benchmarks now emphasize compositional generation and semantic fidelity. GenEval (Ghosh et al., 2023) and T2I-CompBench (Huang et al., 2023) evaluate attributes like object color, counting, and spatial relationships. DPG-Bench (Hu et al., 2024) focuses on generation from dense prompts, and OneIG-Bench (Chang et al., 2025) supports multilingual and stylized evaluation. For image editing, where models modify images according to textual instructions, InstructPix2Pix (Brooks et al., 2023) and MagicBrush (Zhang et al., 2023) assess the correctness of local and global edits. AnyEdit (Yu et al., 2025) introduces a dedicated evaluation set covering 25 editing types for real-world applicability, while ImgEdit (Ye et al., 2025) provides a large-scale benchmark with 1.2 million high-quality edit pairs, emphasizing instruction adherence and visual realism.

## 3. Unison

In this work, we propose Unison, a comprehensive benchmark designed to holistically assess the unified capabilities of UMMs, focusing on the synergy between understanding and generation. To facilitate systematic evaluation, we decompose unified capability into four dimensions: *internal consistency*, *understanding-guided generation*, *generation-guided understanding*, and *mutual enhancement*.

### 3.1. Internal Consistency

Internal consistency, referring to the alignment of UMM's semantic encoding of understanding and generation within its internal representations, is designed by leveraging fine-grained compositional understanding of images alongside text-to-image tasks. Both two tasks fundamentally require holistic comprehension of all input information. In fine-grained compositional understanding, the model must analyze visual elements such as objects, attributes and spatial relationships, typically achieved through tasks like Visual Question Answering (VQA). On the other hand, text-to-image generation requires instruction-following and compositional reasoning to accurately synthesize images. Without this foundational consistency, the model struggles to perform complex tasks effectively, as errors in basic alignment can propagate and hinder performance.

In particular, given an image-caption pair $(I, C)$, a set of questions $Q = \{q_i^{ic}\}_{i=1}^n$ and a set of attributes $A = \{a_i\}_{i=1}^n$, where attributes contain *object, color, texture, count, light text rendering* and *spatial relation*. The image-caption pair $(I, C)$ is manually aligned, where each attribute $a_i$ explicitly represented in the corresponding question $q_i$. For understanding, UMM perform VQA on image $I$ to get answer $s_i^{und} = \mathcal{M}(I, q_i^{ic})$. For generation, we first prompt UMM to perform text-to-image synthesis to obtain a generated image $I_{t2i} = \mathcal{M}(C)$, then apply a dedicated judge model $\mathcal{J}$ to answer the same question on $I_{t2i}$, yielding $s_i^{gen} = \mathcal{J}(I_{t2i}, q_i^{ic})$. Finally, we compare answers $s_i^{und}$ and

*Table 1.* Benchmark Comparison. MU denotes multi-modal understanding, IG denotes image generation, IE denotes image editing, IC denotes internal consistency, UF denotes unified understanding and generation, Det denotes detection, Seg denotes segmentation, HE denotes hybrid editing, HT denotes hybrid type, RI denotes real image, SI denotes synthetic image.

| Benchmark | Evaluation | | | | | Localization | | Editing | | Scenario | | Multi-Turn | Judge Model | Task Type |
|---|---|---|---|---|---|---|---|---|---|---|---|---|---|---|
| | MU | IG | IE | IC | UF | Det | Seg | HE | HT | RI | SI | | | |
| MMBench (Liu et al., 2024) | ✓ | ✗ | ✗ | ✗ | ✗ | ✗ | ✗ | ✗ | 0 | ✓ | ✗ | ✗ | ✗ | 6 |
| SEED-Bench (Li et al., 2024) | ✓ | ✗ | ✗ | ✗ | ✗ | ✗ | ✗ | ✗ | 0 | ✓ | ✗ | ✗ | ✗ | 6 |
| MMMU (Yue et al., 2024) | ✓ | ✗ | ✗ | ✗ | ✗ | ✗ | ✗ | ✗ | 0 | ✓ | ✓ | ✗ | ✗ | 6 |
| GenEval (Ghosh et al., 2023) | ✗ | ✓ | ✗ | ✗ | ✗ | ✗ | ✗ | ✗ | 0 | ✗ | ✓ | ✗ | ✗ | 1 |
| DPG-Bench (Hu et al., 2024) | ✗ | ✓ | ✗ | ✗ | ✗ | ✗ | ✗ | ✗ | 0 | ✗ | ✓ | ✗ | ✗ | 1 |
| OneIG-Bench-EN (Chang et al., 2025) | ✗ | ✓ | ✗ | ✗ | ✗ | ✗ | ✗ | ✗ | 0 | ✗ | ✓ | ✗ | ✗ | 6 |
| AnyEdit (Yu et al., 2025) | ✗ | ✗ | ✓ | ✗ | ✗ | ✗ | ✗ | ✗ | 0 | ✓ | ✓ | ✗ | ✗ | 5 |
| ImageEdit (Ye et al., 2025) | ✗ | ✗ | ✓ | ✗ | ✗ | ✗ | ✗ | ✓ | 6 | ✓ | ✗ | ✓ | ✓ | 3 |
| **Unison** (ours) | ✓ | ✓ | ✓ | ✓ | ✓ | ✓ | ✓ | ✓ | 11 | ✓ | ✓ | ✓ | ✓ | 14 |

$s_i^{gen}$ for each attribute-aligned question $q_i$ calculating final internal consistency score $Score_{ic}$:

$$Score_{ic} = \sum_{i=1}^{n} f(a_i, s_i^{und}, s_i^{gen}))/n, \quad (1)$$

where $f(a_i, s_i^{und}, s_i^{gen}) \in \{0, 1\}$ returns 1 if $s_i^{und}$ and $s_i^{gen}$ are consistent with respect to attribute $a_i$, and 0 otherwise.

Notably, extreme failure cases can lead to *spurious consistency*, where a UMM may misidentify a critical attribute (e.g., an image-text pair depicts five people, but the model perceives six and subsequently generates an image showing six people). Since consistency derived from a faulty premise is meaningless, our evaluation protocol assigns a score of zero to such cases. Our strict criterion ensures that high scores only represent faithful consistency, effectively penalizing hallucinations or inherent model biases.

### 3.2. Understanding-Guided Generation

In this part, we leverage the reasoning capability of UMM as prior to guide contextually accurate generation for synergy's effect assessment. In short, we prompt the UMM to autonomously reason precise edit parts from complex instructions then executing generation. Based on comparison with direct image manipulation, understanding-guided generation provides a systematic analysis for causal impact of a model's comprehension on its generative fidelity.

Specifically, given an image $I$ and a complex editing instruction $E$. The model first comprehends $E$ and reason about $I$ to output the target region $\mathcal{R}$ via bounding box/segmentation mask. Then obtaining its own localization as conditional input, the model executes the editing operation to get final edited image $I_{ie}$. We average the localization score and image editing score to get final score $Score_{ugg}$:

$$Score_{ugg} = (IoU(\mathcal{R}, \mathcal{R}_{gt}) + \mathcal{J}(I, I_{ie})) / 2, \quad (2)$$

where $\mathcal{R}_{gt}$ is the ground truth target region, $IoU(,)$ returns Intersection over Union score, $\mathcal{J}$ is a MLLM-based rating model to provide scores for image editing.

### 3.3. Generation-Guided Understanding

For this paradigm, we employ visual output as a proxy for complex reasoning, exploring the untapped potential of UMMs. We formalize this as generation-guided understanding, which requires a tight coupling between generative synthesis and the comprehension of implicit structural constraints. For example, given a cub net, determining what color of its top face after it is folded into a cube requires visual imagination, which challenges even human cognition. We reveal that UMMs can leverage their generative priors to simulate these transformations, providing an intuitive visual reference that facilitates the resolution of otherwise intractable reasoning problems.

We formalize this task as visual question answering, where an intermediate image generation step is employed to synthesize relevant visual contexts. Based on an image $I$ (2D spatial, 3D spatial) or description $D$ (complex relation), as illustrated in Figure 2, the task is to select the ground-truth option $O_{gt}$ from a set of four candidates in response to a question $q^{ggu}$. To achieve this, UMM first performs an intermediate generative phase explicitly conditioned on the question's context. Subsequently, the model reasons over the synthesized image to derive the final output. To evaluate this process, we define an understanding score $S_u^{ggu} \in \{0, 1\}$ based on the accuracy of the predicted option. The generation score $S_g^{ggu}$ follows the principles of internal consistency. Finally, the holistic performance is measured by the average score $Score_{ggu} = (S_u^{ggu} + S_g^{ggu})/2$.

### 3.4. Mutual Enhancement

In this part, we assess self-refinement ability on capability-specific tasks of UMMs by alternately motivating them alternately comprehend and regenerate their own outputs, or vice versa, referring to *mutual enhancement*. Specifically, the model first generates an initial output, justifies the coherence between the generated content and the input instruction, and then summarizes a textual operation to produce optimized outputs. This iterative loop tests whether UMM can identify and reinforce its own errors without external supervision.

For clarity, this process follows a four-step loop (up to $R$ rounds): 1. Generation: The model produces an initial textual/visual output from the input instruction. 2. Evaluation: The model acts as an evaluator, judging if the current output satisfies the input instruction. 3. Refinement: If misalignments are found, the model needs to generate a specific refinement instruction to reproduce a new output. Then re-evaluates. 4. Termination: The loop stops when the model judges no further conflicts, or reaches the maximum round.

**Self-Refining Image Editing** extends standard editing by incorporating iterative feedback loops.

Formally, assuming there are at most $R$ execution rounds. Given an initial instruction $T_1 \in \{T_r\}_{r=1}^R$, the model generates an edited image $I_1^{me\_ie} \in \{I_r^{me\_ie}\}_{r=1}^R$. For subsequent rounds $r \in \{2, \ldots, R\}$, the model evaluates if $I_{r-1}^{me\_ie}$ fully aligns with the initial instruction $T_1$, If misaligned, it will be prompted to generate a refinement instruction $T_r$ and a corresponding image $I_r^{me\_ie}$, otherwise, the refinement loop terminates early. Moreover, we define self-refining gain $\Delta S_g$ to quantify the cumulative improvement. This metric represents the absolute increase achieved by final refined image $I_k$ compared to $I_1$. We directly use editing score at $k$ round as final score $Score_{me\_ie}$.

**Self-Refining Multimodal Understanding** focuses on ability to diagnose and rectify cross-modal inconsistencies. Unlike self-refining image editing, this task begins with a pre-existing image-caption pair $(I_1, C_1)$ that contain misalignments. The model must critically recognize these discrepancies to drive the refinement process.

In each round $r$, the model performs a "detect-and-correct" cycle, it first identifies semantic mismatches between the current image $I_r^{me\_mu}$ and the original caption $C_1$, then formulates a corrective instruction $T_r$ to produce a better-aligned image $I_{r+1}^{me\_mu}$. This loop terminates when the model perceives no further conflicts or reaches the maximum $R$ rounds. The final performance $Score_{me\_mu}$ is quantified by measuring the alignment between the model-identified mismatches and human-annotated ground truth. Similar to the image editing task, we define self-refining gain $\Delta S_u$ as the absolute improvement from initial state to final round.

The overall mutual enhancement performance is defined as $Score_{me} = (Score_{me\_ie} + Score_{me\_mu})/2$.

# 4. Dataset & Evaluation

Unison supports understanding, generation, and unified assessment. Data distribution is shown in Figure 2. Section 4.1 details the data structure and compares Unison with existing benchmarks. Section 4.2 defines evaluation metrics. Section 4.3 introduces Unison-Judge, an evaluation model closely aligned with human preference for evaluation.

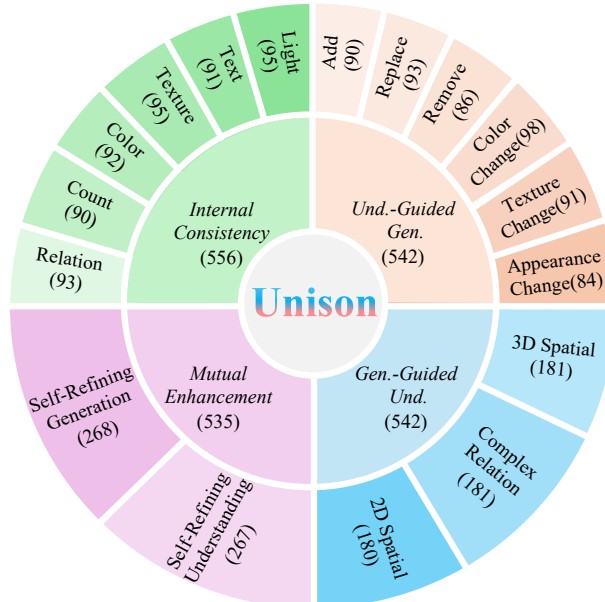

*Figure 2.* Data distribution of Unison. Unison comprises four core categories, each designed to evaluate a specific capability.

## 4.1. Benchmark Construction

To ensure a high-quality, rigorous evaluation framework, our benchmark is constructed through a collaborative pipeline involving state-of-the-art foundation models, followed by three weeks of verification and refinement by ten people.

**Data Corpus and Image Generation**. Our benchmark construction begins with a diverse data curation process tailored to multi-dimensional capabilities. For general vision-language tasks, we derive comprehensive captions from pre-defined attributes following GenEval (Ghosh et al., 2023) protocols. For complex scene understanding, we select a subset from RefCOCOg (Mao et al., 2016) characterized by multi-object clusters and significant occlusions. High-quality aesthetic samples are further sourced from LAION-Aesthetics (Schuhmann et al., 2022) by filtering for scores exceeding 5.0. To generate the visual content, we employ Qwen-Image (Wu et al., 2025a) for text-to-image synthesis, while spatial reasoning images are generated by mapping logical constraints of unfolded diagrams onto 3D structures using OpenCV. For image editing tasks, we define 11 compositional types by combining four meta-operations: *Alter*, *Remove*, *Replace*, and *Add*, ensuring each instruction contains at least two edits to maintain multi-step complexity.

**Instruction Generation and Detection/Segmentation**. To facilitate fine-grained interaction, we implement a multi-stage instruction generation pipeline. We leverage GPT-5.2 (OpenAI, 2025) and Gemini 3 Pro (Team et al., 2023) to produce initial image captions and high-level semantic prompts, which are then refined through customized templates to ensure information integrity. For VQA task

*Table 2.* Comparative results of human alignments between different models. P. denotes parameters.

| Model | P. | Human Alignment ($\rho$) | | | | Average |
|---|---|---|---|---|---|---|
| | | IC | UGG | GGU | ME | |
| Qwen3-VL | 8B | 0.96 | 0.78 | 0.61 | 0.46 | 0.70 |
| GPT-5.2 | - | 0.98 | **0.85** | **0.79** | 0.55 | 0.79 |
| Gemini 3 Pro | - | **0.99** | 0.83 | 0.67 | **0.75** | 0.81 |
| Unison-Judge | 8B | 0.98 | 0.82 | 0.77 | 0.72 | **0.82** |

construction, we utilize the Davidsonian Scene Graph (DSG) (Cho et al., 2023) framework, prompting Qwen3-VL-235B-A22B-Instruct (Bai et al., 2025) to generate complex questions and candidate options. In tasks requiring precise localization, target objects are manually annotated with bounding boxes, which serve as prompts for SAM 3 (Carion et al., 2025) to produce segmentation masks. These masks undergo post-processing to ensure pixel-level accuracy for subsequent editing and understanding evaluations.

**Automated Filtering and Human Annotation**. To guarantee the highest fidelity and logical consistency, all generated data undergoes a rigorous dual-validation process. An initial automated filtering phase is applied to eliminate structural ambiguities and samples that fail to meet aesthetic or alignment thresholds. Subsequently, we conduct an extensive expert manual review. This phase includes: 1) inspecting image-text pairs to ensure all textual attributes are accurately reflected in the visual content; 2) correcting logical inconsistencies in the VQA reasoning chains; and 3) refining instruction prompts to eliminate any potential ambiguities. This refinement process ensures that the benchmark provides high-quality ground truth for evaluating both generative and discriminative capabilities of UMMs.

### 4.2. Evaluation Metrics

We evaluate UMMs using a suite of task-specific metrics. For localization and segmentation, we utilize Intersection over Union (IoU) to quantify predicted regions. For Uni-World, since it generates masks directly on images, we compute pixel-wise alignment by overlaying these masks onto the original images. Image generation is benchmarked against human-annotated VQA data to ensure compositional semantic fidelity. For image editing, we follow ImgEdit (Ye et al., 2025) to employ qualitative ratings on a scale of 0 to 5, which are transformed into a normalized scale of $(0, 1)$, specifically evaluating attribute consistency of edited regions and preservation of unedited areas.

### 4.3. Unison-Judge

To ensure a more reasonable and fine-grained evaluation, we shift from traditional feature-level similarity metrics (Salimans et al., 2016; Heusel et al., 2017; Hessel et al., 2021) to a customized MLLM-based assessment framework, which

aligns more closely with human preference. We introduce Unison-Judge, an evaluation model derived by fine-tuning Qwen3-VL-8B-Instruct on a meticulously curated corpus. This corpus consists of 4,000 manually labeled samples in total, with 1,000 allocated to each dimension to ensure a uniform distribution. Furthermore, we conduct human alignment experiments on 500 samples uniformly sampled across all tasks, as shown in Table 3. Despite Gemini 3 Pro and GPT-5.2 show competitive performance on certain subsets, Unison-Judge achieves the superior average alignment score. This result underscores both the effectiveness and computational efficiency of our proposed framework.

## 5. Experiment

**Experimental Setups.** We evaluate UMMs across a comprehensive suite of both open-source and closed-source platforms. Open-source lineup encompasses Show-o/Show-o2 (Xie et al., 2024; 2025), Janus-pro (Chen et al., 2025b), D-DiT (Li et al., 2025b), ILLUME+ (Huang et al., 2025), OmniGen2 (Wu et al., 2025b), TokenFlow (Qu et al., 2025), Bagel (Deng et al., 2025), SEED-X (Ge et al., 2024), and UniWorld (Lin et al., 2025), respectively. Closed-source models are Gemini 3 Pro (Team et al., 2023) and GPT-5.2 (OpenAI, 2025). To ensure a fair comparison, all models adhere to a standardized input/output format, with image generation and editing resolution fixed at $512 \times 512$. In mutual enhancement experiments, we cap the process at 5 rounds. Notably, early termination is triggered if a model determines that no further errors remain, thereby optimizing computational efficiency.

### 5.1. Evaluation Results

As shown in Table 3, we evaluate various open-source and closed-source models. We averaged the scores from the four dimensions to obtain the final score.

In internal consistency evaluation, we find that understanding performance typically exceeds generation performance. Notably, a model's high comprehension or generation capability does not guarantee good consistency. For example, D-DiT (Li et al., 2025b), despite trailing in absolute scores for both understanding and generation, maintains higher consistency compared to ILLUME+ (Huang et al., 2025). This difference may be attributed to ILLUME+'s hybrid architecture, which decouples understanding and generation, while D-DiT's unified diffusion-based approach enables better cross-capability alignment.

For the Und.-Guided Gen. task, reasoning generally improves contextual understanding. However, IL-LUME+ (Huang et al., 2025) shows a downward trend, with scores dropping from 0.250 to 0.125, due to its low Internal Consistency score, indicating a semantic misalignment that

*Table 3.* Comparative results of open-source and closed-source models on the proposed Unison benchmark. Und., Gen., and Uni. denote understanding, generation, and unified scores, respectively. For open-source models, the bold and underlined values indicate the best and second-best performance within their respective categories. / indicates the model does not support this task.

| Model | Params | Internal Consistency | | | Und.-Guided Gen. | | | Gen-Guided Und. | | | Mutual Enhancement | | | Overall |
|---|---|---|---|---|---|---|---|---|---|---|---|---|---|---|
| | | Und. | Gen. | Uni. | Und. | Gen. | Uni. | Und. | Gen. | Uni. | Und. | Gen. | Uni. | |
| *Open-Source Unified Multimodal Models* | | | | | | | | | | | | | | |
| Show-o | 1.3B | 0.759 | 0.617 | 0.566 | 0.000 | / | / | 0.013 | / | / | / | / | / | - |
| Janus-Pro | 1.5B | 0.883 | 0.694 | 0.679 | 0.002 | / | / | 0.092 | / | / | / | / | / | - |
| Show-o2 | 1.5B | 0.882 | 0.654 | 0.557 | 0.001 | / | / | 0.010 | / | / | / | / | / | - |
| D-DiT | 2B | 0.734 | 0.595 | 0.565 | 0.000 | / | / | 0.005 | / | / | / | / | / | - |
| ILLUME+ | 3B | 0.737 | 0.178 | 0.155 | 0.002 | 0.056 | 0.029 | 0.101 | 0.193 | 0.147 | 0.090 | 0.022 | 0.056 | 0.097 |
| Janus-Pro | 7B | 0.953 | 0.759 | 0.730 | 0.008 | / | / | 0.140 | / | / | / | / | / | - |
| Show-o2 | 7B | **0.971** | 0.771 | 0.758 | 0.000 | / | / | 0.021 | / | / | / | / | / | - |
| ILLUME+ | 7B | 0.796 | 0.221 | 0.177 | 0.005 | 0.083 | 0.037 | 0.140 | 0.250 | 0.125 | 0.005 | 0.113 | 0.059 | 0.113 |
| OmniGen2 | 7B | 0.919 | 0.803 | 0.751 | 0.040 | 0.427 | 0.219 | 0.141 | **0.412** | **0.206** | 0.393 | 0.372 | **0.382** | 0.389 |
| TokenFlow | 14B | 0.928 | 0.541 | 0.505 | 0.000 | / | / | 0.190 | / | / | / | / | / | - |
| BAGEL | 14B | 0.957 | **0.849** | **0.820** | 0.000 | **0.701** | **0.337** | **0.299** | 0.374 | 0.187 | 0.040 | **0.459** | 0.249 | **0.398** |
| SEED-X | 17B | 0.809 | 0.424 | 0.367 | 0.042 | 0.153 | 0.081 | 0.150 | 0.248 | 0.124 | 0.011 | 0.159 | 0.085 | 0.164 |
| UniWorld | 19B | 0.921 | 0.702 | 0.664 | **0.068** | 0.343 | 0.185 | 0.280 | 0.315 | 0.160 | **0.469** | 0.082 | 0.275 | 0.318 |
| *Closed-Source Models* | | | | | | | | | | | | | | |
| Gemini 3 Pro | - | 0.995 | 0.891 | 0.879 | 0.271 | 0.362 | 0.317 | 0.538 | 0.312 | 0.425 | 0.557 | 0.496 | 0.527 | 0.537 |
| GPT-5.2 | - | 0.997 | 0.883 | 0.862 | 0.231 | 0.371 | 0.301 | 0.421 | 0.359 | 0.390 | 0.485 | 0.572 | 0.529 | 0.521 |

limits reasoning effectiveness.

In the Gen-Guided Und. task, the generated content scores remain low, revealing numerous inaccuracies, as shown in Figure 3. However, these errors paradoxically enhance the model's understanding performance, by providing contrastive boundaries that sharpen semantic comprehension.

Finally, in the Mutual Enhancement task, models like Omnigen2 and UniWorld show significant performance gains, with scores of 0.382 and 0.275, respectively. These results suggest that large-scale, integrated models benefit from multi-modal synergies, where proficiency in one modality enhances performance in others.

# 6. Insights and Discussions

**Importance of Internal Consistency.** The results in Table Table 3 highlight that internal consistency is a fundamental prerequisite for high-performing UMMs. We observe a strong correlation between a model's overall score and its performance in the internal consistency dimension. For instance, closed-source models like Gemini 3 Pro (Team et al., 2023) and GPT-5.2 (OpenAI, 2025) exhibit near perfect consistency score, which serves as the foundation for their superior reasoning capabilities. Among open-source models, UniWorld (Lin et al., 2025) and OmniGen2 (Wu et al., 2025b) achieve the highest overall scores (0.398 and 0.376 respectively) by maintaining robust consistency across both understanding and generation tasks. This emphasizes the importance of a UMM's ability to maintain self-alignment, which plays a critical role in minimizing errors.

**UMM's understanding and generation capabilities can synergistic work.** Contrary to the belief that understanding and generation are disparate tasks, our benchmark demonstrates a clear synergistic effect. In the Understanding-Guided Generation and Generation-Guided Understanding,

models do not merely perform well in isolation, they leverage one capability to bootstrap the other. For example, BAGEL achieves a state-of-the-art open-source score of 0.337 in the UGG task. This indicates that the latent representations learned during pre-training provide rich structural priors that enhance understanding, and vice versa.

**UMMs can self-reinforce autonomously.** A pivotal finding from the mutual enhancement category is that UMMs possess the latent ability for autonomous self-improvement. Even when the reference information provided to the model is sub-optimal or contains noise, as implied by the gap between guided and unified scores, models like Omnigen2 and UniWorld show significant gains in the Uni. score (0.382 and 0.275) of Mutual Enhancement. This suggests that these models can perform internal cross-verification, where the generation process acts as a check on understanding, and the understanding module filters generative outputs. This mechanism allows the model to extract value even from incorrect or noisy internal states, effectively achieving a degree of self-reinforcement without external supervision.

# 7. Limitations

**Computational Cost of Evaluation**. Assessing mutual enhancement involves iterative cycles of generation followed by comprehension, and vice versa. This recursive process significantly increases computational overhead, presenting scalability challenges when extending the benchmark to high-resolution imagery or multi-frame video sequences.

**Metric Sensitivity**. Although we introduce Unison-Judge, demonstrating competitive human alignment compared to Gemini 3 Pro, the models under assessment may still exhibit sensitivity to prompt engineering or generative stochasticity. This can introduce a degree of variance that might not fully reflect the model's inherent reasoning limits.

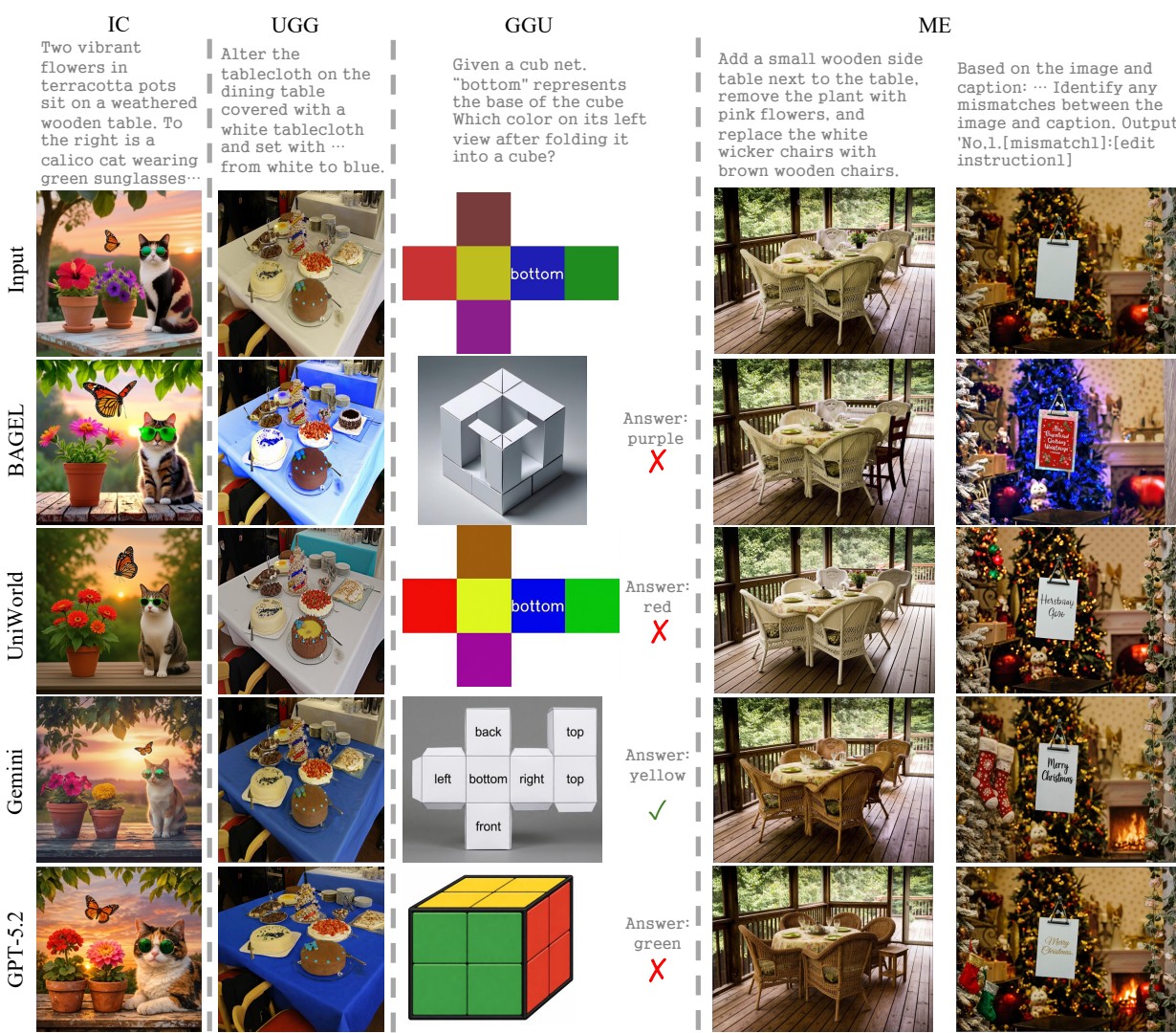

*Figure 3.* Qualitative Visualizations. IC, UGG, GGU and ME denotes Internal Consistency, Understanding-Guided Generation, Generation-Guided Understanding and Mutual Enhancement, respectively. For GGU, visualizations on inclue the 3D spatial task.

## 8. Future Directions

**Inernal Consistency Improvement**. Future research should move beyond standard pre-training objectives by incorporating explicit consistency objective. By penalizing discrepancies between a model's discriminative understanding and its generative output during training, we can foster more robust self-alignment and minimize the hallucinations and error accumulations inherent in current UMMs.

**Self-Refinement Training via Agentic Planning**. A promising direction lies in integrating UMM self-refinement with autonomous agentic planning strategies. By enabling models to autonomously design task-specific plans, they can treat understanding and generation as iterative feedback steps in a self-play loop. This allows the model to generate its own pseudo-supervision signals, facilitating continuous self-evolution and reinforcement learning

**Additional Modality Support**. While current open-source UMMs largely lack video synthesis capabilities, integrating the video modality is essential. Future work should bridge the gap between video understanding and generation, enabling models to simulate physical world dynamics and improve long-range causal reasoning.

## 9. Conclusion

We introduce Unison, a benchmark designed to evaluate the synergistic understanding and generation capabilities of unified multimodal models. Unlike prior benchmarks that assess these skills in isolation, Unison integrates four key dimensions: internal consistency, understanding-guided generation, generation-guided understanding, and mutual

enhancement, and offers both unified and decoupled evaluation tracks for fine-grained analysis. To ensure reliable scoring, we train Unison-Judge, a human-aligned evaluator. Our evaluations reveal critical gaps in current models and demonstrate performance comparable to Gemini 3 Pro.

## Impact Statement

Unison provides a comprehensive framework for unified multimodal research by systematically evaluating the essential synergy between understanding and generation. This work contributes to the reliability of AI systems by identifying internal semantic conflicts and hallucinations, while enabling efficient, human-aligned automated assessment through Unison-Judge. By revealing critical performance gaps in current models, Unison offers practical observations that can help researchers design and develop more consistent and robust unified multimodal architectures in the future.

## Acknowledgment

This work was supported in part by the Science and Technology Commission of Shanghai Municipality under Grant No. 25511103600.

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
