# OpenReview forum: "Unison: Benchmarking Unified Multimodal Models via Synergistic Understanding and Generation"
_ICML.cc/2026/Conference — ICML 2026 regular_

### Official Review · Reviewer_rSj7 · 2026-03-03

**Soundness:** 2
**Presentation:** 2
**Significance:** 3
**Originality:** 2
**Overall Recommendation:** 4
**Confidence:** 4

**Summary:**

The paper introduces Unison, a human-validated benchmark of 2,169 “unified” multimodal task instances intended to measure synergy between vision-language understanding and image generation/editing. It defines four capability dimensions (internal consistency, understanding-guided generation, generation-guided understanding, mutual enhancement), provides unified and decoupled tracks, and proposes an automatic evaluator (Unison-Judge) fine-tuned from Qwen3-VL-8B-Instruct that is reported to correlate with human ratings. Experiments across multiple unified multimodal models suggest that strong performance on isolated understanding or generation tasks does not reliably predict unified-task performance, and that internal consistency is strongly associated with the overall Unison score.

**Compliance With Llm Reviewing Policy:**

Affirmed.

**Final Justification:**

Thanks the authors for addressing my concerns. I'm raising my score to 4.

**Key Questions For Authors:**

Judge robustness: How do Unison-Judge scores change under prompt paraphrases, different reasoning instruction styles, and multiple random seeds (for both judge decoding and model outputs)? If rankings are stable, soundness improves; if not, the benchmark may not support reliable comparisons.

Construct validity: What evidence shows the four dimensions measure distinct phenomena rather than overlapping “judge preference” signals (e.g., factor analysis, low inter-dimension correlations after controlling for overall quality, targeted counterexamples)? Strong construct validity would strengthen originality and significance; weak separation would undermine the taxonomy.

Data/creation bias and leakage: Which foundation models were used in data generation and validation, and do evaluated models overlap with those used to create prompts/captions/edits? Do rankings change when removing samples likely influenced by a given model family? Evidence of leakage would materially reduce confidence in the reported conclusions.

**Limitations:**

no

Add a dedicated “Evaluator Risks” subsection quantifying judge sensitivity (prompt/seed) and providing confidence intervals for model scores and rank stability.

Include stronger human evaluation: stratified per-dimension human ratings, inter-annotator agreement, and disagreement analysis versus the judge to identify systematic judge errors.

Clarify data provenance and anti-leakage controls (model usage during construction; filtering to avoid stylistic fingerprints; ablations removing suspicious subsets).

**Strengths And Weaknesses:**

Soundness: The benchmark’s core claims hinge on an MLLM judge scoring MLLM outputs, creating a “model-evaluates-model” loop that is vulnerable to correlated failure modes, prompt artifacts, and reward hacking. The paper reports correlation with humans but does not convincingly establish judge robustness, weakening confidence in rankings.

Presentation: The taxonomy is intuitive, but key operational details are underspecified: exact prompts, judge training data/rubric, sampling/filtering criteria, and variance reporting are insufficient for faithful replication. The unified/decoupled tracks are useful diagnostically, yet the mapping between tasks, metrics, and success criteria sometimes feels under-justified rather than empirically validated.

Significance: The problem (evaluating unified multimodal systems) is important.

Originality: The synergy-focused framing and four-dimension decomposition are a clear conceptual contribution. However, much of the apparent novelty is in packaging and scoring rather than in demonstrably new measurement validity: without stronger evidence that the dimensions correspond to distinct, reliable constructs, originality feels more taxonomic than methodological.

---

> ### Author Rebuttal · Authors · 2026-03-31
>
> We sincerely appreciate your insightful feedback on our work.
> >**Q1 & Q4 & Q7: MLLM judge paradigm. Unison-Judge robustness. Add Evaluator Risks subsection.**
>
> 1.MLLM judge paradigm.
>
> The entire benchmarking process does not rely entirely on Unison-Judge. For localization and segmentation, we utilize IoU to quantify predicted regions. Image generation is benchmarked against human-annotated VQA data to ensure compositional semantic fidelity. For image editing, we follow ImgEdit. Thus, Unison-Judge is proposed to balance efficiency with evaluation reliability.
>
> 2.Unison-Judge robustness
>
> See Reviewer RPmG Q2 & Q6 due to space limits. Thank you.
>
> 3.Add Evaluator Risks subsection.
>
> We will add a relevant Evaluation Risks subsection in our revised version.
>
>
> >**Q2: Operational details about prompts, training data for Unison-Judge, evaluation metric, data construction criteria, and results variance**
>
> 1.Prompts details.
>
> For image generation, we use:
> "Generate an image based on the {CAPTION}."
>
> For question generation, we use:
> "Task: given input prompts, describe ...
> output format: id | tuple",
> "Task: given input prompts and tuples, describe the parent tuples of each tuple.
> output format: id | dependencies (comma separated)",
> "Task: given input prompts and skill-specific tuples, rewrite each tuple in a natural language question.
> output format: id | question".
>
> For image editing, we use:
> "Task: given an input image, describe ...
>
> Output format:
> id | tuple",
> "Task: given the input image and tuples, describe the parent tuples of each tuple.
>
> Output format:
> id | dependencies (comma separated)",
> "Task: given the input image, tuples and dependencies...
>
> REQUIRED OPERATION: You MUST use every operation "{operation}" for this image. The operation should be clear, actionable.
>
> The edit operations should:
> - Include specific details about what to change (e.g., material, color, text, contents, position)
> - For "Replace" operations, specify what to replace and what to replace it with
> - For "Add" operations, specify what to add and where
> - For "Remove" operations, clearly identify what to remove
> - For "Alter" operations, specify what aspect to alter and how
>
> Output format:
> operation | target object | bbox (comma separated)",
> "Task: given the input image and operations ... hybrid image editing task.
> The instruction should be clear, actionable.
>
> Do NOT generate any extra operations outside input operations for instruction.
>
> Output format:
> instruction".
>
> For caption generation, we use:
> "Task: Analyze the input image and perform a precise captioning task. Operations to Include: {operation}
>
> The caption should:
> * Describe all objects in the Operations.
> * Focus exclusively on target, color...
> * Be strictly limited to 50 words or fewer.
> * Be concise, factual, and avoid unnecessary flowery language.
>
> Output format:
> caption".
>
>
> 2.Evaluation metric, data construction criteria
>
> This can be found in Section 4 of our paper.
>
> 3.Unison-Judge settings and training data
>
> Trained for 10 epochs on 8 A800 GPUs. During finetuning, vision and MLP modules are frozen. LLM is finetuned via LoRA (r=8, α=16, dropout=0.0). Training data can be found in Section 4.3 of our paper.
>
> 4.Results variance
>
> See Reviewer P2xH Q4.1 & Q9. Thank you.
>
> >**Q3 & Q5: The four dimensions measure evidence of different abilities.**
>
> We argue that the four dimensions are essential and complementary. Collectively, they establish a holistic evaluation framework for the current UMM. Experiments can be found at Reviewer P2xH Q1. Thank you.
>
> >**Q6 & Q9: Data construction bias regarding model overlap.**
>
> See Reviewer P2xH Q3.2 & Q7. Thank you.
>
>
> >**Q8: Additional human evaluations.**
>
> We provide additional statistics based on our human annotations during benchmark construction. When the Spearman Correlation $\rho$ between understanding and generating individual ability scores is taken into account, Unison-Judge still achieves the best human consistency score.
>
> |Model|IC-Und.|IC-Gen. |IC-Uni. |UGG-Und. |UGG-Gen. |UGG-Uni. |GGU-Und. |GGU-Gen. |GGU-Uni. |ME-Und. |ME-Gen. |ME-Uni. |Average |
> |---|---|---|---|---|---|---|---|---|---|---|---|---|---|
> |Qwen3-VL|0.95|0.96|0.96|0.84|0.71|0.78|0.70|0.55|0.61|0.52|0.44|0.46|0.71|
> |GPT-5.2|0.98|0.96|0.98|0.88|0.82|0.85|0.82|0.81|0.79|0.61|0.52|0.55|0.80|
> |Gemini 3 Pro|1.00|0.97|0.99|0.90|0.77|0.83|0.74|0.66|0.67|0.69|0.74|0.75|0.81|
> |Unison-Judge|0.98|0.97|0.98|0.89|0.78|0.82|0.81|0.73|0.77|0.70|0.70|0.72|0.82|
>
> Additionally, we provide Spearman Correlation between all annotators and the judge, as shown below.
>
> |Annotator|Spearman Correlation $\rho$|
> |---|---|
> |A1|0.9864|
> |A2|0.9911|
> |A3|0.9872|
> |A4|0.9925|
> |A5|0.9896|
> |A6|0.9905|
> |A7|0.9887|
>
> Our multi-round annotation workflow and correlation analysis demonstrate that our dataset is highly consistent with human preferences, minimizing bias in the data.

---

> > ### Author Rebuttal · Reviewer_rSj7 · 2026-04-02
> >
> > I notice other reviewers have rightly questioned the judge's architectural bias and prompt robustness. However, a more fundamental issue remains: Unison-Judge is built upon an 8B parameter model. Yet, it is tasked with evaluating the outputs of vastly more capable models like GPT-5.2, Gemini 3 Pro, and 10B+ open-source UMMs.
> > In complex reasoning or highly compositional generation tasks, an 8B model may lack the intrinsic capacity to recognize the nuanced correctness of a much larger model, leading to a 'blind leading the sighted' scenario.
> > Did the authors conduct any error analysis on instances where Unison-Judge penalized GPT-5.2 or Gemini 3 Pro? How many of these penalties were actual errors by the evaluated models, versus the 8B judge simply failing to comprehend a correct but complex output?

---

> > > ### Author Response · Authors · 2026-04-03
> > >
> > > Dear Reviewer rSj7,
> > >
> > > Thank you for your thoughtful review and acknowledgment. We sincerely appreciate your constructive feedback, and the replies are as follows:
> > >
> > > >**QI: An 8B parameters model may lack the intrinsic capacity to recognize the nuanced correctness of a much larger model.**
> > >
> > > We agree and acknowledge that an 8B parameters model may have limitations in complex reasoning or compositional generation compared to larger-scale models. However, **Unison-Judge is designed specifically for discriminative evaluation and only applies to the Unison benchmark we proposed**. Our analysis is as follows:
> > >
> > > 1.Statistics analysis.
> > >
> > > We conduct experiments comparing Unison-Judge with other larger-scale open-source models for human alignment. As shown below:
> > >
> > > | Judge Model | IC ($\rho$) | UGG ($\rho$) | GGU ($\rho$) | ME ($\rho$) | Average |
> > > |---|---|---|---|---|---|
> > > | InternVL3-38B | 0.97 | 0.78 | 0.66 | 0.44 | 0.71 |
> > > | InternVL3-78B | 0.98 | 0.78 | 0.63 | 0.49 | 0.72 |
> > > | Qwen3-VL-30B-A3B-Instruct | 0.96 | 0.80 | 0.63 | 0.48 | 0.72 |
> > > | Qwen3-VL-235B-A22B-Instruct | 0.97 | 0.81 | 0.66 | 0.51 | 0.74 |
> > > | Unison-Judge-8B | 0.98 | 0.82 | 0.77 | 0.72 | 0.82 |
> > >
> > > Although InternVL3-78B has a larger number of parameters than InternVL3-38B, its human alignments on the GGU task are lower (0.63<0.66). InternVL3-78B has significantly fewer parameters than Qwen3-VL-235B-A22B-Instruct, but its human consistency on the IC task is higher (0.98>0.97). **The statistics demonstrate that LLM's judging capabilities cannot be concluded solely based on the size of its parameters.** Unison-Judge has been task-specifically fine-tuned based on our benchmark and offers a significant advantage over larger-scale open-source models targeting general tasks.
> > >
> > >
> > > 2.Related works about small-scale LLMs as Judges.
> > >
> > > Zhu et al. [1] propose JudgeLM, demonstrating that **strategically fine-tuning 7B/13B LLMs can achieve evaluation performance competitive with GPT-4, and even exhibit superior human alignments in domain-specific tasks**. Additionally, Imgedit [2] fine-tunes Qwen2.5-VL-**7B** to get Imgedit-Judge for evaluating complex image editing. OneIG-Bench [3] also utilizes Qwen2.5-VL-**7B** to evaluate large-scale models such as GPT-4o.
> > >
> > >
> > > **In conclusion, our empirical results and previous research indicate that a judge model with superior evaluation capability does not need to match the complex reasoning or generation performance of the evaluated models, and often requires fewer parameters.**
> > >
> > > [1] JudgeLM: Fine-tuned Large Language Models are Scalable Judges, Lianghui Zhu et al., ICLR 2025.
> > >
> > > [2] Imgedit: A unified image editing dataset and benchmark, Ye, Yang et al., NeurIPS 2025.
> > >
> > > [3] OneIG-Bench: Omni-dimensional Nuanced Evaluation for Image Generation, Chang, Jingjing et al., NeurIPS 2025.
> > >
> > >
> > >
> > > >**QII: Error analysis of GPT-5.2 and Gemini 3 Pro on misunderstanding correct output versus actual errors.**
> > >
> > > For GPT-5.2 and Gemini 3 Pro, we select all evaluation results from the Unison-Judge evaluation pipeline for statistical error analysis. In the notation X/Y: X (values to the left of "/") denotes the number of samples where correct outputs are fully or partially recognized as errors by the Unison-Judge. Y (values to the right of "/") denotes the total number of samples annotated by humans as fully or partially errors. "-" indicates that Unison-Judge is not used in the corresponding dimension evaluation. As shown below:
> > >
> > > |Model|IC-Und.|IC-Gen.|IC-Uni.|UGG-Und.|UGG-Gen.|UGG-Uni.|GGU-Und.|GGU-Gen.|GGU-Uni.|ME-Und.|ME-Gen.|ME-Uni.|Average Misjudgement Rate
> > > |---|---|---|---|---|---|---|---|---|---|---|---|---|---
> > > | Gemini 3 Pro | - | 2/28 | 1/42 |-| 11/229 | 2/264 | - | 37/487 | 7/522 | 31/237 | 25/269 | 30/497 | **5.83%**
> > > | GPT-5.2 | - | 1/21 | 0/37 |-| 12/224 | 3/248 | - | 26/456 | 5/475 | 30/276 | 46/310 | 24/492 | **5.41%**
> > >
> > > Statistical analysis demonstrates that Unison-Judge reliably identifies errors in frontier models with a misjudgment rate of less than 6%. Notably, the misjudgment rates for GPT-5.2 and Gemini 3 Pro are remarkably consistent, with a variance of less than 0.5%. **This stability highlights the model-agnostic robustness of Unison-Judge**.
> > >
> > > We hope our response resolves your issue. Thank you for your time and participation.
> > >
> > > Best regards,
> > > Authors of Paper #567

---

### Official Review · Reviewer_aCAQ · 2026-03-04

**Soundness:** 3
**Presentation:** 3
**Significance:** 3
**Originality:** 3
**Overall Recommendation:** 4
**Confidence:** 4

**Summary:**

This paper introduces Unison, a comprehensive benchmark designed to evaluate Unified Multimodal Models (UMMs) through the lens of synergistic understanding and generation. The manuscript deals with the inadequacy of current evaluation methods that treat understanding and generation as decoupled tasks, thereby failing to capture the full potential of natively unified architectures. The main contributions of the work include a multi-dimensional synergistic framework and a high-quality dataset and evaluator. The experimental results highlight that while leading models like GPT-5.2 and Gemini 3 Pro show strong performance, current open-source UMMs still face significant challenges in maintaining internal consistency and performing autonomous self-reinforcement.

**Compliance With Llm Reviewing Policy:**

Affirmed.

**Final Justification:**

The rebuttal clearly reinforced my prior assessment. I've decided to maintain my scores.

**Key Questions For Authors:**

1. In Section 3.1, you mention that your evaluation protocol assigns a score of zero to cases where a model is "consistently wrong" (e.g., misidentifying an attribute in understanding and then generating based on that same error). If this process is automated via Unison-Judge, how do you ensure the judge does not mistake a model's "consistent hallucination" for genuine $Score_{ic}$?

2. In the experimental analysis (Section 5.1), you observe that even inaccurate generated content can "paradoxically enhance" a model’s understanding by providing "contrastive boundaries." Could you provide a deeper qualitative or quantitative analysis of how "wrong" images assist in "right" reasoning? Specifically, is the model performing a form of "visual cot" and "thinking-with-image," or are the errors simply acting as noise that forces the model to rely more heavily on the textual prompt?

3. The Mutual Enhancement (ME) task involves up to 5 rounds of alternating generation and understanding. In your experiments, did you observe instances of "semantic drift," where the model’s corrections in round $r$ led to new, unrelated errors in round $r+1$? Detailed statistics on whether the $Score_{me}$ generally plateaus or fluctuates across the 5 rounds would provide insight into the stability of UMM self-refinement and whether more iterations are truly beneficial or lead to divergence.

**Limitations:**

yes

**Strengths And Weaknesses:**

Strengths:
- The decomposition of "synergy" into four distinct tracks—Internal Consistency, UGG, GGU, and ME—is theoretically sound. It covers the full lifecycle of information flow in a unified model (input-to-output and output-to-input).
- The authors develop Unison-Judge, which demonstrates high correlation with human judgment and significantly strengthens the validity of the results.
- Most existing benchmarks treat understanding and generation as two separate leaderboards. Unison’s originality lies in its focus on the interplay between these capabilities.
- The concept of evaluating a model's ability to "self-correct" by looking at its own generated images and then refining its understanding is a creative and novel way to measure emergent properties of unified models.

Weaknesses:
- The mathematical formulation of the "Mutual Enhancement" (ME) track is somewhat dense. A more intuitive walkthrough of the iterative feedback loop in the main text would improve accessibility for readers.
- There are some problems in writing, e.g., Line 381 "Inernal" -> "Internal".

---

> ### Author Rebuttal · Authors · 2026-03-31
>
> We sincerely appreciate your insightful feedback on our work: synergistic workflow, Unison-Judge, good originality, and novel concept.
> >**Q1: More intuitive demonstration for Mutual Enhancement.**
>
> For clarity, the Mutual Enhancement process follows a four-step loop (up to $R$ rounds):
> 1. Generation: The model produces an initial textual/visual output from the input instruction.
> 2. Evaluation: The model acts as an evaluator, judging if the current output satisfies the input instruction.
> 3. Refinement: If misalignments are found, the model needs to generate a specific refinement instruction to reproduce a new output. Then re-evaluates.
> 4. Termination: The loop stops when the model judges no further conflicts, or reaches the maximum round $R$.
>
> We will include the above explanation in the paper.
>
> >**Q2: Minor typo errors in writing.**
>
> Thank you for carefully reading our work. We will revise it accurately.
>
> >**Q3: Details on judging spurious consistency in Internal Consistency.**
>
> In Internal Consistency (IC), the overall assessment pipeline is partially automated by Unison-Judge. Generation capabilities are evaluated by Unison-Judge, while understanding capabilities are self-evaluated by the UMM itself. Details are as follows:
>
> Our evaluations are based on Visual Question Answering (VQA) data, and every question asks only a single attribute, such as *"Are there 9 pencils?"*, *"Is the cup white?"*, etc. For attribute $a_i$ in an image-caption pair $(I, C)$, we use Unison-Judge to evaluate if the image $I_{t2i}$ generated by a UMM correctly represents $a_i$.
>
> Regarding generation, we input question $q_i^{ic}$ and generated image $I_{t2i}$ to Unison-Judge, and prompt it to answer *"yes"* or *"no"*. $s_i^{gen} \in \{0,1\}$ returns 1 if Unison-Judge answers *"yes"*, otherwise 0.
>
> For understanding, we input question $q_i^{ic}$ and image $I$ which is from the original image-caption pair to a UMM, and prompt it to answer *"yes"* or *"no"*. $s_i^{und} \in \{0,1\}$ returns 1 if the UMM answers *"yes"*, otherwise 0. **We claim that this part is designed to rigorously verify UMMs' true comprehension abilities. Since the image-caption pair and question are manually aligned, the model only needs to understand whether a certain attribute exists, allowing for a direct evaluation.**
>
> For final score calculation, $Score_{ic}=1$ only when both $s_i^{gen}$ and $s_i^{und}$ are equal to 1, otherwise 0. For question *"Are there 9 pencils?"*, we assume that a UMM generates an image representing 8 pencils, and it misunderstands that the original image also has 8 pencils. In such case, $s_i^{gen}=0$ and $s_i^{und}=0$, resulting in $Score_{ic}=0$. Therefore, our evaluation pipeline will not mistake a model's consistent hallucination.
>
>
> >**Q4: Quantitative analysis for paradoxical enhancement in Generation-Guided Understanding.**
>
>
> For each Gen. score interval, we compute the proportion of samples for which the model produces correct answers (Accuracy), based on the count of samples within that interval.
>
> |Model|Accuracy (Gen. 0~0.03)|Accuracy (Gen. 0.03~0.06)|Accuracy (Gen. 0.06~0.09)| Accuracy (Gen. 0.09~0.12)|Accuracy (Gen.>0.12)|GGU Uni.|
> |---|---|---|---|---|---|---|
> |SEED-X|36/431=0.084|79/111=0.712|0/0|0/0|0/0|0.213|
> |ILLUME+ 3B|40/429=0.093|68/93=0.731|16/20=0.800|0/0|0/0|0.224|
> |ILLUME+ 7B|41/424=0.097|72/96=0.750|20/22=0.909|0/0|0/0|0.233|
> |UniWorld|39/399=0.098|72/112=0.643|28/36=0.778|4/5=0.80|0/0|0.265|
> |OmniGen2|36/383=0.094|58/85=0.682|40/47=0.851|23/27=0.852|0/0|0.290|
> |BAGEL|31/374=0.083|67/81=0.827|45/48=0.938|27/30=0.90|0/0|0.314|
> |GPT-5.2|3/171=0.018|74/130=0.569|51/91=0.560|82/116=0.707|42/45=0.933|0.464|
> |Gemini 3 Pro|10/203=0.049|130/151=0.861| 100/120=0.833|67/68=0.985|0/0|0.567|
>
> As shown in the table above, model accuracy positively correlates with the generated Gen. score: lower scores correspond to reduced accuracy, while higher scores indicate stronger alignment between the generated image and the instruction semantics. It can be concluded that the generated images contribute positively to comprehension, even in the presence of generation errors or semantic deviations.
>
> >**Q5: Semantic drift and stability across different round numbers in Mutual Enhancement.**
>
> We conduct experiments on ME subset data. We sort all samples in ME according to the average Uni. scores, and uniformly select 100 samples from them. We force the model to infer 10 rounds, even if it determines that the current result is perfect. Results are shown in [Convergence Analysis](https://anonymous.4open.science/r/icml-F1C6/ME.png), which reveal that after 5 rounds, the performance of all models remain basically unchanged except for BAGEL. Therefore, based on your valuable insight, a UMM can improve within a reasonable number of rounds, but semantic drift may occur when the number of rounds is too large. And we will add the analysis of this issue in a later version of our paper.

---

> > ### Author Rebuttal · Reviewer_aCAQ · 2026-04-02
> >
> > Thank you for your rebuttal. I've decided to maintain my scores.

---

> > > ### Author Response · Authors · 2026-04-02
> > >
> > > Dear Reviewer aCAQ,
> > >
> > > Thank you for your thoughtful review and positive acknowledgment. We sincerely appreciate your constructive feedback, which has helped us improve the clarity and quality of our paper. We will carefully incorporate your suggestions in the final revision.
> > >
> > > Best regards,
> > > Authors of Paper #567

---

### Official Review · Reviewer_RPmG · 2026-03-12

**Soundness:** 3
**Presentation:** 3
**Significance:** 3
**Originality:** 2
**Overall Recommendation:** 4
**Confidence:** 3

**Summary:**

The authors present Unison, a novel benchmark designed to evaluate Unified Multimodal Models (UMMs) not just on isolated tasks, but on the synergistic interplay between their comprehension and generative capabilities. The paper introduces a dataset of 2,169 human-validated samples distributed across four distinct evaluation dimensions: 1. Measures whether a model's generated visual output aligns with its own semantic comprehension of a given input. 2. Understanding-Guided Generation (UGG): Evaluates the model's ability to use its reasoning and spatial localization as a prior for accurate image editing/generation. 3. Generation-Guided Understanding (GGU): Tests whether the model can synthesize intermediate visual contexts (e.g., imagining a folded cube net) to solve complex spatial reasoning questions. 4. Mutual Enhancement (ME): Assesses the model's capacity for autonomous self-refinement through iterative cycles of generation and comprehension.

**Compliance With Llm Reviewing Policy:**

Affirmed.

**Final Justification:**

The authors have resolved most of my concerns. However, the further questions are not replied, and it is appoarching to the deadline, so I will keep my previous score.

**Key Questions For Authors:**

**Questions:**

* 1. Could the authors discuss how models utilizing flow matching might theoretically perform on the Internal Consistency (IC) and Mutual Enhancement (ME) dimensions compared to standard diffusion or autoregressive models? Have you conducted any preliminary evaluations on such architectures?

* 2. Have the authors quantified the potential for architectural bias within Unison-Judge? For instance, what is the agreement rate between Unison-Judge and human annotators specifically when evaluating Qwen-based models versus architecturally distinct models (like D-DiT or SEED-X)?

* 3. Can the authors provide a brief convergence analysis (e.g., graphing $\Delta S_g$ across rounds 1 through 10 on a representative subset of data) to empirically justify the 5-round limit? Do models typically plateau before round 5, or is there evidence of continued enhancement that the benchmark currently misses?

**Limitations:**

Yes

**Strengths And Weaknesses:**

**Strengths:**

* 1. The transition from decoupled assessment to synergistic evaluation is a critical and necessary step for the field. The introduction of GGU is particularly insightful; explicitly probing whether a multimodal foundation model can use its generative capacity as a "visual scratchpad" for complex spatial reasoning is an elegant way to quantify the benefits of unified architectures.

* 2. The formalization of Internal Consistency is highly robust. By assigning a score of zero to instances where the model exhibits "spurious consistency" (generating an image based on a faulty initial perception), the benchmark effectively penalizes hallucination and ensures that high scores reflect genuine multimodal alignment.

* 3. The ME evaluation dimension provides a valuable, quantifiable metric for a UMM's autonomous self-play capabilities. Demonstrating that models like UniWorld can achieve measurable gains from internal cross-verification—even from noisy internal states—is a strong empirical contribution.


**Weakness:**

* 1. The paper categorizes the generative mechanisms of current UMMs strictly into auto-regressive, diffusion-based, and hybrid models. For a benchmark aiming to evaluate state-of-the-art unified multimodal foundation models, this taxonomy is slightly constrained. It overlooks rapidly emerging continuous-time generative frameworks, specifically those utilizing flow matching. Because flow matching constructs continuous vector fields and often operates over joint latent spaces differently than standard diffusion, evaluating how these architectures behave regarding Internal Consistency (IC) or Mutual Enhancement (ME) would significantly elevate the benchmark's longevity and relevance to next-generation generative models.

* 2. The Unison-Judge evaluator is fine-tuned from Qwen3-VL-8B-Instruct. However, Qwen models (specifically the Qwen3-VL family) are heavily featured as baselines or foundational architectures for other models in the evaluation. The authors must address the potential for "judge bias"—the phenomenon where an LLM-as-a-judge disproportionately favors outputs generated by its own base architecture or tokenization strategy.

* 3. The authors rightly acknowledge the computational cost of evaluating the Mutual Enhancement (ME) dimension due to its iterative nature. However, capping the self-refinement at 5 rounds might artificially truncate the self-correction curves of larger models. Providing a convergence analysis (e.g., graphing $\Delta S_g$ across rounds 1 through 10 on a subset of data) would better justify this hyperparameter choice.

* 4. Lack of insightful analysis.

---

> ### Author Rebuttal · Authors · 2026-03-31
>
> We are thankful and very encouraged by your positive comments.
> >**Q1 & Q5: Additional evaluation and discussion for flow matching-based UMMs.**
>
>
> We leverage FLUX.1 (FLUX.1 Kontext) [1], Qwen-Image [2], and SD 3.5 [3] (Stable Diffusion 3.5 Large Turbo) to conduct additional evaluation on Unison benchmark. Since these models do not support textual output, we use InternVL3.5 [4] (InternVL3.5-241B-A28B) as the understanding model.
>
>
> | Model | IC-Uni. | UGG-Uni. | GGU-Uni. | ME-Uni. | Overall |
> |---|---|---|---|---|---|
> | Qwen-Image + InternVL3.5 | 0.880 | 0.721 | 0.373 | 0.422 | 0.599 |
> | SD 3.5 + InternVL3.5 | 0.912 | 0.749 | 0.433 | 0.495 | 0.647 |
> | FLUX.1 + InternVL3.5 | 0.894 | 0.764 | 0.468 | 0.510 | 0.659 |
>
> The table provides strong evidence that FLUX.1 is currently the leading model among SOTA flow matching-based generative models, particularly in handling complex semantic understanding and editing tasks. Meanwhile, SD 3.5 remains the top choice for basic generation tasks due to its extreme stability. A pivotal direction for future UMM research involves significantly enhancing complex reasoning capabilities while maintaining high generation quality, following the paradigm established by FLUX.1.
>
>
> [1] Flux. 1 kontext: Flow matching for in-context image generation and editing in latent space, Batifol et al., 2025
>
> [2] Qwen-image technical report, Wu, Chenfei et al., 2025
>
> [3] Stable Diffusion 3.5, Stability AI, 2025
>
> [4] InternVL3.5: Advancing Open-Source Multimodal Models in Versatility, Reasoning, and Efficiency, Weiyun Wang et al., 2025
>
> >**Q2 & Q6: Judge bias analysis for Unison-Judge.**
>
> Among UMMs in our evaluations, Show-o2, ILLUME+, TokenFlow, BAGEL, OmniGen2, and Uniworld both adopt Qwen2.5 series as their understanding or visual encoder module. SEED-X adopts the visual encoder from Qwen-VL. VILA-U, show-o, Janus-Pro, and D-DiT use distinct architectures.
>
> We conduct experiments for Qwen-based/distinct architectures on the agreement rate (Spearman Correlation $\rho$) between Unison-Judge and human, as shown below:
>
> | Model | IC ($\rho$) | UGG ($\rho$) | GGU ($\rho$) | ME ($\rho$) | Average |
> |---|---|---|---|---|---|
> | Show-o | 0.997 | 0.823 | 0.762 | 0.712 | 0.824 |
> | Janus-Pro | 0.985 | 0.811 | 0.771 | 0.741 | 0.827 |
> | D-DiT | 0.984 | 0.819 | 0.775 | 0.723 | 0.825 |
> | VILA-U | 0.981 | 0.829 | 0.789 | 0.715 | 0.829 |
> | Show-o2 | 0.980 | 0.804 | 0.741 | 0.735 | 0.815 |
> | ILLUME+ | 0.984 | 0.817 | 0.752 | 0.744 | 0.824 |
> | TokenFlow | 0.981 | 0.823 | 0.739 | 0.724 | 0.817 |
> | BAGEL | 0.978 | 0.837 | 0.748 | 0.721 | 0.821 |
> | OmniGen2 | 0.975 | 0.796 | 0.781 | 0.722 | 0.819 |
> | SEED-X | 0.971 | 0.801 | 0.780 | 0.716 | 0.817 |
> | Uniworld | 0.979 | 0.814 | 0.781 | 0.715 | 0.822 |
>
> From the table above, we can obtain the average agreement rate of distinct series is 0.826 (first four lines), Qwen series is 0.819 (last seven lines). This indicates that the bias of using the Qwen3-VL structure is negligible when evaluating Qwen2.5-based models versus architecturally distinct models.
>
>
> >**Q3 & Q7: Convergence analysis for Mutual Enhancement.**
>
> We sort all samples in ME according to the average Uni. scores, and uniformly select 100 samples from them. We force the model to infer 10 rounds, even if it determines that the current result is perfect. Convergence results are shown in [Convergence Analysis](https://anonymous.4open.science/r/icml-F1C6/ME.png), which show that after 5 rounds, the performance of all models remain basically unchanged except for BAGEL. This revels that a UMM can improve within a reasonable number of rounds, but semantic drift may occur when the number of rounds is too large.
>
> We will add the analysis of this issue in a later version of our paper.
>
>
>
> >**Q4: Insightful analysis.**
>
> Based on your insightful comments and our additional experiments in Q1 & Q3 & Q5 & Q7, we reveal that:
>
> 1.There remains a notable performance gap in current open-source UMMs. Concatenating models with strong understanding and generative capabilities can serve as a high-performance alternative, but this counters high inference costs;
>
> 2.Flow matching-based methods (e.g., FLUX.1) show stronger performance in tasks requiring both understanding and generation (GGU/ME), while traditional diffusion models (e.g., SD 3.5) offer more stable results for basic generation tasks. This suggests that the choice of generation approach significantly influences a UMM's overall capabilities. Future designs could benefit from task-specific adaptations or hybrid approaches that combine the strengths of different methods;
>
> 3.UMMs can improve within a reasonable number of rounds, but semantic drift may occur when the number of rounds is too large. Using reinforcement learning to post-train the current UMM by leveraging its synergistic effect is a worth exploring research direction.

---

> > ### Author Rebuttal · Reviewer_RPmG · 2026-04-03
> >
> > Thanks, the authors have resolved most of my concerns. However, the further questions are not replied, and it is appoarching to the deadline, so I will keep my previous score.

---

> > > ### Author Response · Authors · 2026-04-03
> > >
> > > Dear Reviewer RPmG,
> > >
> > > Thank you for your thoughtful review and positive acknowledgment. We sincerely appreciate your constructive feedback, which has helped us improve the clarity and quality of our paper. We will carefully incorporate your suggestions in the final revision.
> > >
> > > Best regards,
> > > Authors of Paper #567

---

### Official Review · Reviewer_P2xH · 2026-03-12

**Soundness:** 2
**Presentation:** 2
**Significance:** 3
**Originality:** 3
**Overall Recommendation:** 4
**Confidence:** 4

**Summary:**

This paper introduces Unison, a benchmark for evaluating unified multimodal models (UMMs) through tasks that jointly involve understanding and generation, rather than assessing the two capabilities separately. The benchmark is organized into four dimensions: internal consistency, understanding-guided generation, generation-guided understanding, and mutual enhancement, and contains 2,169 human-validated samples. The paper also presents Unison-Judge, an automatic evaluator obtained by fine-tuning Qwen3-VL-8B-Instruct on 4,000 manually labeled samples and validating on 500 human-alignment examples. Experiments compare multiple open-source and closed-source UMMs, with Gemini 3 Pro and GPT-5.2 obtaining the highest overall scores in Table 3, while the benchmark is used to study how current UMMs perform on tasks that require interaction between understanding and generation.

**Compliance With Llm Reviewing Policy:**

Affirmed.

**Final Justification:**

My final recommendation is Weak Accept. The paper addresses a relevant and reasonably original benchmark problem for unified multimodal models, and its benchmark design and empirical coverage are meaningful. My main concerns were about the strength of some broader claims, robustness to scoring choices, comparability across models, and reproducibility details. The rebuttal did not fully resolve every issue, but it provided useful additional quantitative analysis and robustness evidence, which increased my confidence in the soundness of the work and led me to adjust my score upward from Weak Reject to Weak Accept.

**Key Questions For Authors:**

- Section 6 makes several broader claims about internal consistency, synergistic effects, and autonomous self-improvement. Could the authors provide additional quantitative support for these claims, such as controlled ablations, correlation statistics, or significance analysis? A stronger response here would improve my assessment of soundness.

- How sensitive are the benchmark rankings to the chosen scoring protocol, especially the equal weighting in UGG/GGU, the strict zeroing rule in internal consistency, and the scoring rule used in mutual enhancement? If the rankings remain stable under reasonable alternatives, that would increase confidence in the robustness of the benchmark.

- Since GPT-5.2 and Gemini 3 Pro are used in the benchmark construction pipeline and also evaluated as baselines, did the authors examine whether this introduces any style or distribution bias? Additional analysis on this point could clarify the interpretation of the closed-source results.

- Table 3 includes models with incomplete task support. Would the main conclusions change if the comparison were restricted to models that support all four task categories? A clarified analysis here would make the cross-model comparison easier to interpret.

- The limitations section mentions prompt sensitivity and generative stochasticity. Could the authors report multi-run variance or confidence intervals for the main generation-heavy tasks? This would help readers assess how much weight to place on close score differences.

**Limitations:**

Partially. The paper discusses computational cost and metric sensitivity, and it includes a societal impact section. One additional limitation that could be discussed more explicitly is the possible effect of using GPT-5.2 and Gemini 3 Pro in the benchmark construction pipeline while also evaluating them as baselines.

**Strengths And Weaknesses:**

**Strengths**

* The paper studies a relevant evaluation setting for unified multimodal models (UMMs): rather than testing understanding and generation separately, it evaluates their interaction through four dimensions—internal consistency, understanding-guided generation, generation-guided understanding, and mutual enhancement. This gives the benchmark a clearly defined scope.

* The benchmark design is structured in a way that supports diagnostic analysis. In addition to unified evaluation, the paper also considers decoupled evaluation, and Table 1 helps clarify how Unison differs from prior understanding, generation, and editing benchmarks in terms of task coverage and evaluation granularity.

* The empirical section covers a reasonably broad set of models for a benchmark paper. The paper evaluates both open-source and closed-source UMMs under a standardized setup, and also introduces Unison-Judge as an automatic evaluator, which achieves the highest average human alignment among the compared judges in Table 2.

**Weaknesses**

* Some of the broader claims in Section 6 appear to rely mainly on descriptive result patterns rather than stronger supporting analysis. For example, statements about internal consistency as a prerequisite, clear synergistic effects, or autonomous self-improvement would be better supported with additional quantitative analysis or controlled ablations.

* The paper does not provide much analysis of how sensitive the benchmark outcomes are to specific protocol choices, such as equal weighting in UGG/GGU, strict zeroing in internal consistency, or the use of final-round scores in mutual enhancement. As a result, it is somewhat difficult to assess how robust the reported rankings are to alternative but reasonable metric definitions.

* There are some comparability questions in the experimental setup. Table 3 includes models that do not support all task categories, which makes cross-model interpretation less direct. In addition, GPT-5.2 and Gemini 3 Pro are used in the benchmark construction pipeline and later evaluated as baselines, which may warrant additional discussion or analysis regarding possible benchmark-construction bias.

* The paper would be easier to assess and reproduce with more detail on robustness and implementation. Although the limitations section notes prompt sensitivity and generative stochasticity, the experiments do not report variance, confidence intervals, or multi-run results; similarly, more detail on prompts, filtering thresholds, and judge-training settings would improve reproducibility.

---

> ### Author Rebuttal · Authors · 2026-03-31
>
> We sincerely appreciate your insightful feedback on our paper.
>
> >**Q1 & Q5: Quantitative analysis and ablations for Section 6.**
>
> **1. Internal Consistency (IC) as a prerequisite**
>
> We conduct Spearman Correlation analysis between each IC sub-metric and the overall score across eight models supporting all unified tasks. The Und. and Gen. metric shows a weaker correlation, indicating that single capability alone has a limited contribution to holistic performance.
>
>
> |IC Metrics|UGG Uni. ($\rho$)|GGU Uni. ($\rho$)|ME Uni. ($\rho$)|
> |---|---|---|---|
> |Und.|0.461|0.587|0.498|
> |Gen.|0.637|0.667|0.732|
> |Uni.|0.941|0.965|0.927|
>
> **2. UMMs' synergistic effects**
>
> For the UGG task, we provide the Ground Truth of the understanding part to the model's input. Similarly for the GGU task.
>
> |Models|UGG-Gen.|Uni. (GT Input)|$\Delta$ (%)|GGU-Und.|Uni. (GT Input)|$\Delta$ (%)|
> |---|---|---|---|---|---|---|
> |ILLUME+ 3B|0.212|0.253|2.5|0.202|0.286|6.4|
> |ILLUME+ 7B|0.238|0.306|1.7|0.221|0.290|5.7|
> |OmniGen2|0.382|0.413|0.8|0.262|0.401|11.1|
> |BAGEL|0.365|0.398|1.7|0.282|0.410|9.6|
> |SEED-X|0.185|0.215|2.8|0.185|0.315|10.2|
> |UniWorld|0.345|0.425|1.1|0.252|0.377|11.2|
> |Gemini 3 Pro|0.762|0.797|0.3|0.538|0.779|21.2|
> |GPT-5.2|0.771|0.819|0.4|0.421|0.643|17.9|
>
>
> **3. UMMs' autonomous self-improvement**
>
> Due to the length limit, please see the answers of Reviewer RPmG Q3 & Q7 and Reviewer aCAQ Q5. Thank you.
>
>
> >**Q2 & Q6: Sensitivity to protocol choices.**
>
> 1.Alternative for UGG/GGU
>
> We use Geometric Mean to recalculate Uni. scores for all 8 models that support UGG/GGU task. $S_{ugg}^{u}$, $S_{ugg}^{g}$ represent Und. and Gen. scores in UGG, then $Score_{ugg} = \sqrt{S_{ugg}^{u} \times S_{ugg}^{g}}$. Similarly, $Score_{ggu} = \sqrt{S_{ggu}^{g} \times S_{ggu}^{u}}$.
>
> |Model|UGG-Uni.|GGU-Uni.|Overall|
> |---|---|---|---|
> |SEED-X|0.179|0.222|0.250|
> |ILLUME+ 7B|0.292|0.240|0.331|
> |BAGEL|0.390|0.308|0.376|
> |OmniGen2|0.403|0.299|0.383|
> |UniWorld|0.421|0.259|0.399|
> |GPT-5.2|0.813|0.470|0.696|
> |Gemini 3 Pro|0.803|0.557|0.708|
>
>
> 2.Alternative for internal consistency
>
> We use a more fine-grained score range for the $Score_{ic}$, where $f(a_i,s_i^{und},s_i^{gen}) \in \{0, 0.5, 1\}$. We leverage Unison-Judge to evaluate whether a UMM's understanding is consistent with its generated image. $f(,)$ returns 0.5 if a consistency is achieved, otherwise returns 0.
>
> With the participation of Unison-Judge, only 4 models exhibit this situation, as shown below:
>
> |Model|IC-Uni.|
> |---|---|
> |VILA-U|0.6235->0.6238|
> |Show-o2 7B|0.6491->0.6497|
> |UniWorld |0.7240->0.7251|
> |OmniGen2|0.8106->0.8112|
>
> 3.Alternative scoring rule used in mutual enhancement
>
> We calculate the average Uni. score across all 5 rounds, as shown below:
>
> |Model|ME Uni. (Average)|ME Uni. (Last Round)|
> |---|---|---|
> |SEED-X|0.026|0.025|
> |ILLUME+ 7B|0.142|0.144|
> |OmniGen2|0.143|0.148|
> |UniWorld|0.167|0.189|
> |BAGEL|0.176|0.208|
> |GPT-5.2|0.455|0.529|
> |Gemini 3 Pro|0.459|0.527|
>
> >**Q3.1 & 8: Discussion for models which support all four task categories.**
>
> The main conclusions have been further analyzed and validated in the answers to Q1 & Q5, so they will not change.
>
>
> >**Q3.2 & Q7: Analysis on distribution bias for closed-source results.**
>
> 1.Human Verification Impact
>
> We compare initial model-generated and final human-verified samples: average sentence cosine similarity is 0.21, indicating substantial human revision, with 64.04% (17,244/26,925) of tokens added/removed/replaced during verification.
>
>
> 2.Construction Ablation
>
> To analyze potential bias, we use raw ME and UGG data containing key attributes. Samples are ranked by average Uni. scores from Gemini 3 Pro and GPT-5.2, then 100 per dataset are selected to form ME-100 and UGG-100. GPT-4o rewrites these with substituted attributes, yielding ME-100-4o and UGG-100-4o; similarly, Gemini and GPT generate ME-100-G$^2$ and UGG-100-G$^2$.
>
>
> |Models|UGG-100-G$^2$|ME-100-G$^2$|UGG-100-4o|ME-100-4o|
> |---|---|---|---|---|
> |Gemini 3 Pro|0.7726|0.5423|0.7712 (-0.0014)|0.5429 (+0.0006)|
> |GPT-5.2|0.8031|0.5454|0.8038 (+0.0007)|0.5466 (+0.0012)|
>
>
> >**Q4.1 & Q9: Prompt sensitivity and generative stochasticity.**
>
> In our work, comparative results of Table 3 are averaged across 5 different seeds during model inference, the main generation-heavy task Mutual Enhancement's results are shown below:
>
> |Model|ME-Uni.|
> |---|---|
> |ILLUME+ 7B|0.144 (+-0.002)|
> |OmniGen2|0.148 (+-0.001)|
> |BAGEL|0.208 (+-0.001)|
> |SEED-X|0.025 (+-0.004)|
> |UniWorld|0.189 (+-0.002)|
>
>
> We additionally conduct experiments on 5 different prompt templates rewritten by GPT-4o, as shown below:
>
> |Model|P 1|P 2|P 3|P 4|P 5|
> |---|---|---|---|---|---|
> |ILLUME+ 7B|0.144|0.145|0.145|0.144|0.144|
> |OmniGen2|0.148|0.146|0.147|0.148|0.148|
> |BAGEL|0.208|0.210|0.209|0.210|0.209|
> |SEED-X|0.025|0.025|0.027|0.025|0.026|
> |UniWorld|0.189|0.189|0.189|0.190|0.189|
>
>
> >**Q4.2: Details on prompts, filtering thresholds, and judge-training settings.**
>
> See Reviewer rSj7 Q2 due to space limits. Thank you.

---

> > ### Author Rebuttal · Reviewer_P2xH · 2026-04-02
> >
> > I appreciate the authors' clarifications and additional analyses in the rebuttal, which partially address my concerns.

---

> > > ### Author Response · Authors · 2026-04-03
> > >
> > > Dear Reviewer P2xH,
> > >
> > > Thank you for your thoughtful review and positive acknowledgment. We sincerely appreciate your constructive feedback, which has helped us improve the clarity and quality of our paper. We will carefully incorporate your suggestions in the final revision.
> > >
> > > Best regards,
> > > Authors of Paper #567

---

### Decision · Program_Chairs · 2026-04-30

**Decision:**

Accept (regular)

**Comment:**

This paper introduces a benchmark for evaluating unified multimodal models on joint understanding and generation, addressing the limitation of prior evaluations that treat the two capabilities separately. It further includes a learned evaluator aligned with human judgments. Experiments reveal clear gaps in jointly modeling understanding and generation, highlighting key challenges and future research directions.

It received review comments from four reviewers. Before the rebuttal, a series of concerns were raised. Detailed feedback was provided by the authors. Most of them were addressed well. The AC reviewed the paper, peer review comments, and author responses. The AC considered the research question addressed in the paper to be important, and although its technical contributions were relatively weak, it still provided sufficient insights into the research field and can inspire future works. Therefore, the AC recommended acceptance.